# Enhanced piezoelectricity from highly polarizable oriented amorphous fractions in biaxially oriented poly(vinylidene fluoride) with pure β crystals

Yanfei Huang [1,8], Guanchun Rui [2,8], Qiong Li[2], Elshad Allahyarov [3,4,5], Ruipeng Li[6], Masafumi Fukuto[6], Gan-Ji Zhong[7], Jia-Zhuang Xu[7], Zhong-Ming Li[7], Philip L. Taylor[3✉] & Lei Zhu [2✉]

Piezoelectric polymers hold great potential for various electromechanical applications, but only show low performance, with $|d_{33}| < 30$ pC/N. We prepare a highly piezoelectric polymer ($d_{33} = -62$ pC/N) based on a biaxially oriented poly(vinylidene fluoride) (BOPVDF, crystallinity = 0.52). After unidirectional poling, macroscopically aligned samples with pure β crystals are achieved, which show a high spontaneous polarization ($P_s$) of 140 mC/m$^2$. Given the theoretical limit of $P_{s,\beta} = 188$ mC/m$^2$ for the neat β crystal, the high $P_s$ cannot be explained by the crystalline-amorphous two-phase model (i.e., $P_{s,\beta} = 270$ mC/m$^2$). Instead, we deduce that a significant amount (at least 0.25) of an oriented amorphous fraction (OAF) must be present between these two phases. Experimental data suggest that the mobile OAF resulted in the negative and high $d_{33}$ for the poled BOPVDF. The plausibility of this conclusion is supported by molecular dynamics simulations.

[1] College of Materials Science and Engineering, Shenzhen Key Laboratory of Polymer Science and Technology, Shenzhen University, 518055 Shenzhen, PR China. [2] Department of Macromolecular Science and Engineering, Case Western Reserve University, Cleveland, OH 44106-7202, USA. [3] Department of Physics, Case Western Reserve University, Cleveland, OH 44106, USA. [4] Theoretische Chemie, Universität Duisburg-Essen, D-45141 Essen, Germany. [5] Theoretical Department, Joint Institute for High Temperatures, Russian Academy of Sciences, 13/19 Izhorskaya street, Moscow 125412, Russia. [6] National Synchrotron Light Source II, Brookhaven National Laboratory, Upton, New York 11973, USA. [7] College of Polymer Science and Engineering, State Key Laboratory of Polymer Materials Engineering, Sichuan University, 610065 Chengdu, Sichuan, PR China. [8]These authors contributed equally: Yanfei Huang, Guanchun Rui. ✉email: plt@case.edu; lxz121@case.edu

**B**ulk polymer piezoelectrics exhibit a performance inferior to that of ceramic piezoelectrics, especially those with a morphotropic phase boundary (MPB) behavior[1]. For example, most PVDF-based polymers possess the highest reported piezoelectric coefficients ($d_{33}$ and $d_{31}$) with absolute values usually <30 pC/N (or pm/V)[2,3], whereas lead zirconate titanate (PZT) has a $d_{33}$ as high as ca. 550 pC/N[1]. To enable polymers for practical piezoelectric applications, it is highly desirable to further enhance their piezoelectric performance. Surprisingly, despite decades of research into piezoelectric polymers, the physical mechanism for reversible polymer piezoelectricity is still a matter of debate. Basically, this debate is centered on the question: which component is primarily responsible for the observed piezoelectricity: the crystal ($d_{33}$[3–5]), the amorphous phase [$d_{33}$ (the dimensional model)[6–10] and $d_{31}$[4,11]], or the crystal-amorphous interface?[12,13] Without a clear fundamental understanding of the underlying origin, it is difficult to further enhance the piezoelectric performance for polymers.

On the basis of a previous analysis[9], a physical basis for $d_{3j}$ has been established for PVDF-based piezoelectrics in the form of the following relation:

$$d_{3j} = P_3 \left( \frac{\partial \ln M_3}{\partial T_j} - \frac{\partial s_3}{\partial s_j} J_j \right) \quad (1)$$

where $P_3$ is the polarization ($P_3 = P_{r0} + \Delta P$; $P_{r0}$ is the permanent remanent polarization and $\Delta P$ is the stress-induced polarization), $M_3$ the macroscopic dipole moment, $T_j$ the applied stress ($j = 1$ or 3), $s_j$ the strain (negative for $s_3$ and positive for $s_1$), and $J_j$ the elastic compliance (negative for $J_3$ and positive for $J_1$). From Eq. 1, there are several factors that increase $d_{3j}$. First, the $P_{r0}$ is important for piezoelectricity, and this is why PVDF-based polymers need to be macroscopically polarized to exhibit piezoelectricity. Second, the stress-induced $\Delta P$ is also important and can enhance $d_{3j}$ through the first term in the parentheses of Eq. 1 ($M_3 = P_3 V$, where $V$ is the sample volume). It is usually high when the polymer dielectric constant is high. Third, the compliance $J_j$ is insignificant for ceramics, but important for polymers due to their lower modulus. Herein, we discuss how the crystals and the amorphous phase affect these parameters for PVDF-based polymers.

First, ferroelectric PVDF crystals are essential to achieve a high and stable $P_{r0}$ via high-field poling. In this sense, the ferroelectric β phase is the most desired because of its high spontaneous polarization (ca. 188 mC/m²)[14,15]. Crystal orientation is often employed to maximize $P_{r0}$. However, high-quality PVDF films with pure β crystals capable of surviving high-field poling are difficult to obtain. As a consequence, P(VDF-*co*-

trifluoroethylene) [P(VDF-TrFE)] random copolymers have been pursued. When the TrFE content is above 20 mol.%, the Curie temperature ($T_C$) is accessible below the melting temperature[16]. Below $T_C$, a pure ferroelectric phase is always obtained, regardless of the film processing method used. Usually, the rigid PVDF crystals do not contribute significantly to the change of $M_3$ under low electric fields. However, recent reports have shown that an MPB-like behavior was achieved for P(VDF-TrFE) around the 50/50 composition, where a field-induced conformation transformation from a $3_1$ helix to a zig-zag enhanced the piezoelectric performance[3,5]. Nevertheless, this mechanism still does not explain the reasonably good piezoelectric performance of neat PVDF or P(VDF-TrFE) with a high VDF content.

Second, the isotropic amorphous phase (IAF) in PVDF-based polymers should not exhibit any piezoelectric property. However, it provides the high compliance in Eq. 1[9]. Nevertheless, this dimensional model cannot fully explain the high $d_{31}$ for PVDF. Recently, the coupling between the ferroelectric crystal and the amorphous phase at their interfaces was proposed to explain the high piezoelectric performance and negative $d_{33}$ for P(VDF-TrFE) 65/35[13]. However, this coupling effect is yet to be fully clarified.

In this study, we prepare a highly piezoelectric polymer with $d_{33}$ as high as −62 pC/N based on a poled biaxially oriented PVDF (BOPVDF) film containing pure β crystals. The fundamental mechanism behind the high piezoelectricity is revealed by structural characterization and computer simulation, and is attributed to the presence of a large amount of oriented amorphous fraction (OAF) in BOPVDF that renders a high spontaneous polarization ($P_s$ ~ 140 mC/m²). This understanding provides the guidance for further improving the piezoelectricity of ferroelectric polymers.

## Results and discussion

**Structure characterization of the poled BOPVDF.** Pure β-phase crystals for a BOPVDF film (8.0 μm thick) were achieved by repeated unidirectional poling at 650 MV/m (10 Hz) for at least 40 times. After poling, all the ferroelectric domains in the crystals were aligned in the normal direction (ND) of the film. The crystalline structure of the poled BOPVDF was characterized by two-dimensional (2D) edge-on wide-angle X-ray diffraction (WAXD) and Fourier transform infrared (FTIR) spectroscopy experiments. The one-dimensional (1D) WAXD profiles in Fig. 1a were obtained from 2D WAXD patterns. Major reflections for the α and β crystals are labeled in Fig. 1a. The flat-on WAXD results are shown in Supplementary Fig. 1 in the Supplementary Information. For the fresh BOPVDF film, both α and β crystals

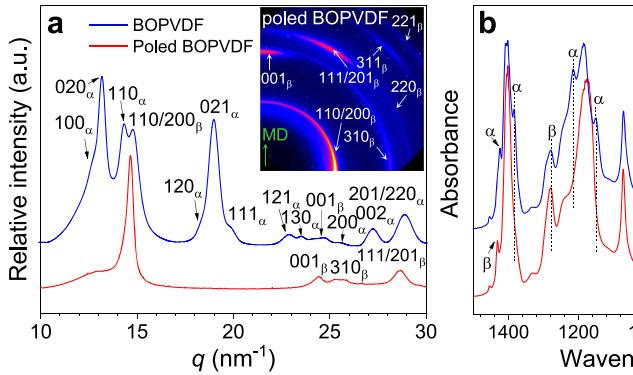
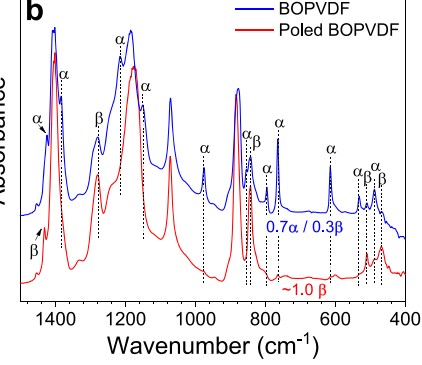

**Fig. 1 Structural characterization by WAXD and FTIR. a** 1D WAXD profiles for the fresh and poled BOPVDF films at room temperature. The inset shows the 2D WAXD pattern (in a logarithmic scale) for the poled BOPVDF film. The X-ray beam is along the transverse direction (TD) and the machine direction (MD) is vertical. **b** FTIR spectra for the fresh and poled BOPVDF films in the transmission mode. Absorption bands for α and β crystals are labeled.

were observed from the 1D WAXD profile (Fig. 1a; 2D edge-on pattern in Supplementary Fig. 1a) and the FTIR spectrum (Fig. 1b). From a differential scanning calorimetry study, the crystallinity was estimated to be ~0.54 (weight fraction)[17]. From the quantitative FTIR analysis (Fig. 1b)[17], the α and β phases in the PVDF crystals were determined to be 70% and 30%, respectively.

For the highly poled BOPVDF film, only the β crystal reflections were observed, as seen in Fig. 1a: $(110/200)_\beta$, $(310)_\beta$, $(220)_\beta$, $(001)_\beta$, $(201)_\beta$, $(311)_\beta$, and $(221)_\beta$. The β crystallinity ($f_\beta$) was estimated from the WAXD profile to be 0.52 (see Supplementary Fig. 2), suggesting that the high-field electric poling did not change the overall crystallinity, but transformed all α crystals into β crystals. This conjecture was supported by the FTIR analysis in Fig. 1b, from which all α absorption bands had disappeared. Both the WAXD and FTIR results indicated that pure β-phase crystals were successfully achieved for the BOPVDF film after extensive unipolar poling at 650 MV/m.

**Ferroelectric properties of the poled BOPVDF.** The ferroelectric behavior of these BOPVDF films was studied by electric displacement-electric field (D-E) loop tests, and the results are shown in Fig. 2a, b (AC electronic conduction was subtracted[18], as shown in Supplementary Fig. 3). Broad hysteresis loops were observed for both samples. Surprisingly, the poled BOPVDF film exhibited a much higher spontaneous polarization ($P_{s,film} = 140\ mC/m^2$) than that ($67\ mC/m^2$) of the fresh BOPVDF film when the poling field was 300 MV/m at 10 Hz (Fig. 2a). Meanwhile, the dynamic permittivity [$\kappa = \partial D/\partial(\varepsilon_0 E) = 22.9$] at the high field for the poled BOPVDF film was

significantly higher than that ($\kappa = 9.3$) for the fresh BOPVDF film. Finally, the coercive field ($E_c = 88\ MV/m$) for the poled BOPVDF film was also lower than that ($E_c = 112\ MV/m$) of the fresh BOPVDF film. The dielectric constant of the highly poled BOPVDF was studied by broadband dielectric spectroscopy (BDS) at 25 °C; see the inset of Fig. 2b (the complete BDS results are shown in Supplementary Fig. 4). The real part of the relative dielectric constant ($\varepsilon_r'$) was found to be 19.5 at 10 Hz, which is significantly higher than that ($\varepsilon_r' = 11.5$ at 10 Hz) of the fresh BOPVDF (Supplementary Fig. 4a)[17].

After subtraction of the deformational polarization ($D_{def}$; see Fig. 2b), which was extracted from measurements (see Supplementary Note 3) on polycarbonate (PC)/PVDF multilayer films[19], the nonlinear polarization ($P_{NL}$)-E loops were obtained (see Fig. 2c or d). Given the crystallinity, $f_\beta = 0.52$, the $P_s$ for pure β PVDF crystals ($P_{s,\beta}$) could be estimated. If we assume the two-phase model (i.e., the β crystal and IAF; see the inset in Fig. 2c), $P_{s,\beta} = 270\ mC/m^2$ at 300 MV/m. Compared to the maximum $P_{s,\beta}$ of 188 mC/m² calculated from density functional theory[14,15], which takes into account the coupling interaction between rigid dipoles and electronic cloud, this high value is not possible. Accordingly, a three-phase model should be employed; see the inset of Fig. 2d. From our previous work, we proposed an OAF between the lamellar crystal and the IAF[17]. Basically, chain-folding is not the primary structure at the crystal-amorphous interfaces for highly stretched PVDF samples. Instead, many chains are pulled out from the crystal basal planes by the large-scale plastic deformation, forming the OAF. During the bipolar electric poling, the OAF must participate in the domain formation and subsequent ferroelectric switching. By assuming the maximum $P_s$ for the OAF ($P_{s,OAF}$) to be $P_{s,\beta}$, the OAF content

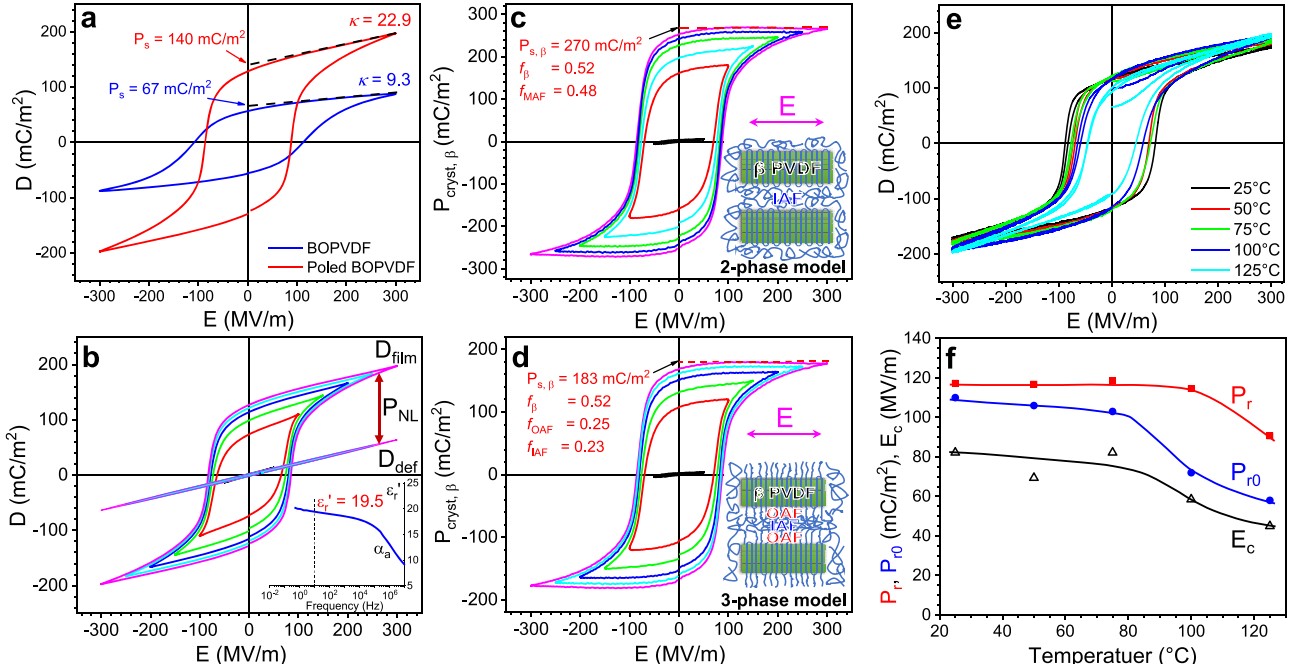

**Fig. 2 Ferroelectric property of the poled BOPVDF. a** Comparison of the bipolar D-E loops for the fresh and poled BOPVDF films at 300 MV/m. **b** Progressive bipolar D-E loops for the poled BOPVDF under different electric fields at room temperature. The poling field has a sinusoidal waveform at 10 Hz. The inset shows the frequency-scan real part of the relative permittivity ($\varepsilon_r'$) at 25 °C. The extracted linear D-E loops from the deformational polarization ($D_{def}$) of the poled BOPVDF film are also shown, following the method in a previous report (see Supplementary Note 3)[19]. After subtracting the $D_{def}$ loop from the bipolar D-E loops for the poled BOPVDF film, nonlinear P-E loops are obtained for **c** the two-phase and **d** the three-phase models. The inset two-phase model in **c** contains the β lamellar crystals and the isotropic amorphous fraction (IAF). The inset three-phase model in **d** contains the β crystals, the IAF, and the oriented amorphous fraction (OAF) connecting the lamellar crystal and the IAF. **e** First two consecutive bipolar D-E loops for the highly poled BOPVDF film after thermal annealing at different temperatures for 30 min. **f** Determination of instantaneous $P_r$, permanent $P_{r0}$, and $E_c$ from the bipolar D-E loops in **e** for the highly poled BOPVDF at different temperatures.

($f_{OAF}$) can be estimated as:

$$P_{s,film} = P_{s,\beta}f_\beta + P_{s,OAF}f_{OAF} \leq P_{s,\beta}\left(f_\beta + f_{OAF}\right) \qquad (2)$$

The minimum value of $f_{OAF}$ is calculated to be 0.25. Based on this study, we conclude that the OAF must exist and participate in the ferroelectric domain formation and subsequent switching upon high-field electric poling.

In the past, the OAF at the PVDF crystal-amorphous interfaces has been inferred from the observation that PVDF/poly(methyl methacrylate) (PMMA) blends exhibited two glass transition temperatures ($T_g$s);[20–22] one of which remained constant around −35 °C, while the other increased with the PMMA content. It is speculated that the OAF is so densely packed at the PVDF crystal-amorphous interfaces that no PMMA chains can penetrate into it. Only the IAF in PVDF is fully miscible with PMMA to exhibit variable $T_g$ depending on the PMMA content. However, the participation of the OAF in the ferroelectric switching has not previously been reported. This is the first account to suggest that the OAF (at least 0.25) directly participates in the ferroelectric switching of PVDF.

From Eq. 1, high piezoelectric performance is expected when the permanent remanent polarization ($P_{r0}$) is high. We note that this $P_{r0}$ is different from the in-situ remanent polarization ($P_r$) during ferroelectric switching under high fields. Because of the relaxation of ferroelectric domains over time after electric poling, $P_{r0} < P_r$, as reported before[23]. To subtract the relaxed polarization from the in-situ $P_r$ and determine the $P_{r0}$ in the highly poled BOPVDF at different temperatures, we designed the following experiment. After 30 min of thermal annealing of the freshly poled BOPVDF film in a silicone oil bath at a preset temperature (25–125 °C), the first two bipolar D-E loops were recorded at 300 MV/m (10 Hz). These bipolar D-E loops at different temperatures are shown in Fig. 2e. From these D-E loops, $P_{r0}$, $P_r$, and $E_c$ were determined according to the method described in Supplementary Note 5 (see Supplementary Fig. 5), and results are shown in Fig. 2f. As we can see, $P_{r0}$ was always lower than $P_r$, and it remained above 100 mC/m$^2$ when the temperature was below 75 °C. It is significant that these permanent $P_{r0}$ values are much higher than those reported for in-situ $P_r$ in the literature (usually $P_r$ = 55–80 mC/m$^2$)[24]. Given such a high $P_{r0}$, high piezoelectric performance could be anticipated for the highly poled BOPVDF film based on Eq. 1.

The in-situ $P_r$ started to decrease above 100 °C, whereas the permanent $P_{r0}$ and $E_c$ started to decrease above 75 °C. The decrease in $P_r$ could be attributed to a decrease of the ferroelectric domain size with increasing temperature. The decrease in $P_{r0}$ indicated the relaxation of aligned ferroelectric domains as a result of enhanced thermal motion in the β crystals at high temperatures. This observation is consistent with our previous report[25]. The decreased $P_{r0}$ above 80 °C would potentially decrease the piezoelectric performance at high temperatures; indeed, this has also been observed in the literature[26].

**Piezoelectric properties of the poled BOPVDF.** Given the high $P_{r0}$ values of the highly poled BOPVDF, the piezoelectric coefficients ($d_{33}$ and $d_{31}/d_{32}$) were measured using direct piezoelectric measurements. Here, $d_{33}$ is the piezoelectric coefficient along the film ND, $d_{31}$ is along the MD, and $d_{32}$ is along the TD. The experimental setups are shown in Supplementary Note 6 (Supplementary Figs. 6, 7). To avoid the possibility of triboelectric charge generation during the direct piezoelectric measurement of $d_{33}$, a gold-plated copper disk (3.2 mm diameter) was placed in direct contact with the gold-coated poled BOPVDF sample. As a control experiment, the same $d_{33}$ measurement was performed for the fresh BOPVDF film, and it was confirmed that no

piezoelectric or triboelectric charge generation was observed (see Supplementary Fig. 8).

The results of direct piezoelectric charge measurements are shown in Supplementary Fig. 9. Using Eqs. 4 and 7 in Methods section, direct piezoelectric coefficients were obtained. As can be seen in Fig. 3a, $d_{33}$ was negative and its absolute value increased upon increasing the applied dynamic stress. Below 0.1 MPa, a normal $|d_{33}|$ ~18 pC/N was obtained. This value was similar to that (~21 pC/N) determined by a $d_{33}$ piezo meter with a fixed dynamic stress of ca. 3.7 kPa and a static force of 2.5 N (see the red star in Fig. 3a). Detailed comparison of these two methods is shown in Supplementary Note 9 (Supplementary Figs. 10–12). Above 0.8 MPa, $|d_{33}|$ reached a plateau value ~62 pC/N. This value is significantly higher than those for conventional PVDF[2,3], and is similar to that reported for P(VDF-TrFE) copolymer near the 50/50 composition[5]. From the $d_{31}$ and $d_{32}$ measurements, we also observed stress-dependent $d_{31}$ and $d_{32}$ (Fig. 3b). Similar stress-dependent piezoelectric coefficients were also reported previously[12]. The maximum values reached were $d_{31}$ = 22 pC/N at 41 MPa and $d_{32}$ = 18 pC/N at 49 MPa, respectively. These values are typical for conventional PVDF homopolymers[2,3]. Because of the biaxial orientation, $d_{31}$ and $d_{32}$ were positive and lower than $|d_{33}|$, consistent with a similar trend reported in the literature[27]. The different trends of stress dependence for $d_{33}$ and $d_{31}/d_{32}$ can be explained by the dimensional effect. Based on Tashiro et al.[4], the dimensional effect is much more significant for $d_{33}$ than $d_{31}/d_{32}$, because the interchain distance is more easily changed than the covalently bonded chain length by an external stress. As a result, it is likely that $d_{33}$ has a stronger dependence on the dynamic stress than $d_{31}/d_{32}$, as observed in our experimental results in Fig. 3.

To determine the electromechanical coupling factor ($k_{3j}$, where $j$ = 1, 2, 3), we carried out modulus ($Y_j$) measurements along various directions. Using the dielectric constant of 19.5, the $k_{3j}$ were calculated using the following equation:

$$k_{3j}^2 = d_{3j}^2 Y_j / \varepsilon_r \varepsilon_0 \qquad (3)$$

For the compression modulus $Y_3$, a nanoindentation experiment was performed. The result is shown in Fig. 3c. From the recovery curve, $Y_3$ was determined to be 4.79 GPa, following a method reported in the literature[28]. The maximum $k_{33}$ was then calculated to be 0.303, which is higher than values commonly reported in the literature[2,3]. From the tensile experiments (Fig. 3d), $Y_1$ and $Y_2$ were determined to be 3.44 and 3.05 GPa, respectively. The maximum $k_{31}$ and $k_{32}$ were consequently calculated to be 0.092 and 0.070, respectively. These values are similar to those reported in the literature for conventional PVDF[2,3].

By taking into account the combined existence of the OAF and the high piezoelectric performance for the highly poled BOPVDF, a mechanism for the piezoelectricity in PVDF is proposed in Fig. 4. From the correlation function analysis of the small-angle X-ray scattering (SAXS) curve for the poled BOPVDF (see Supplementary Fig. 13), the overall lamellar spacing was 11.8 nm. The amorphous layer and crystalline lamellar thicknesses were 6.02 and 5.78 nm, respectively, given that the crystallinity was 0.52. Due to reorganization during the large-scale plastic deformation, a significant portion of the amorphous chains do not form chain folds; instead, they form the OAF. Only in the center of the amorphous layer is the IAF present. Because the β PVDF crystals are highly poled in the upward direction, the OAF adjacent to the crystal-amorphous interfaces must also keep its dipole moment upward. On passing into the interior of the IAF, the upward dipole alignment must gradually randomize. When a dynamic stress is applied, an immediate in-plane strain (δ) is

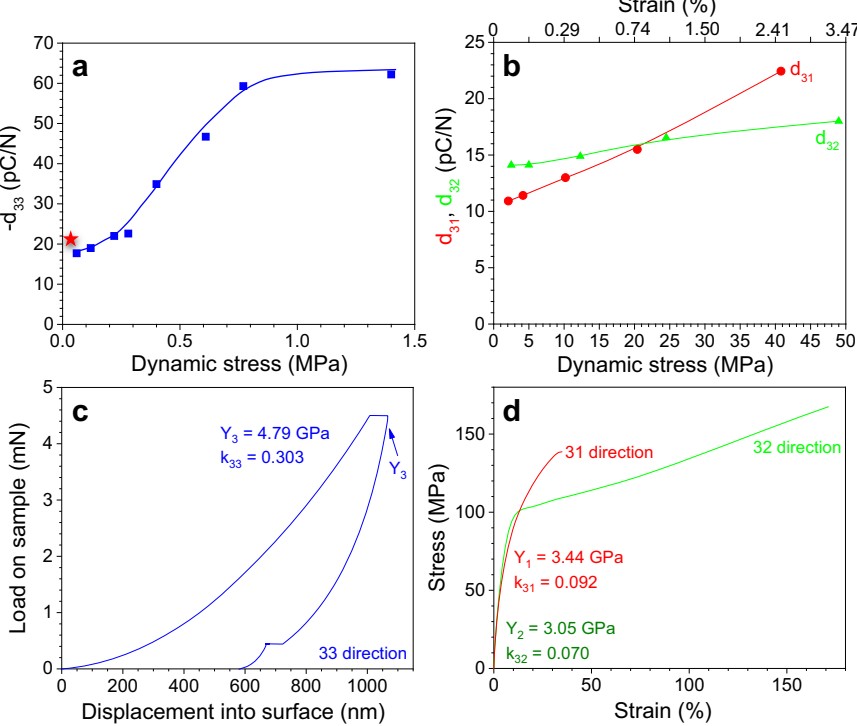

**Fig. 3 Piezoelectric and mechanical properties of the poled BOPVDF.** Using direct piezoelectric measurement, various piezoelectric coefficients are determined: **a** $d_{33}$ and **b** $d_{31}$, $d_{32}$ as a function of dynamic stress for the highly poled BOPVDF film. The red star in **a** indicates the $d_{33}$ value measured by the $d_{33}$ piezo meter with a static force of 2.5 N. **c** Compression modulus ($Y_3$) is determined by nanoindentation, and **d** tensile moduli ($Y_1$ and $Y_2$) are determined by the stress-strain curves for the highly poled BOPVDF film. The maximum $k_{3j}$ ($j = 1$, 2, and 3) values are shown in **c** and **d**.

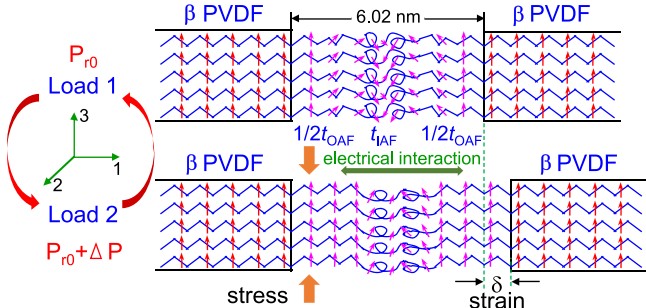

**Fig. 4 Direct piezoelectric effect.** Schematic representation of stress-induced direct piezoelectric effect.

induced. As a result, more dipoles in the mobile OAF will be rotated upward, leading to an increase in polarization, $\Delta P$. In turn, the upward dipoles in the OAF will also induce an electrical repulsion in the film plane, further pushing the crystals apart. This mechano-electrical interaction is actually the reason for the positive $d_{31}/d_{32}$ and the negative $d_{33}$ values observed for PVDF and P(VDF-TrFE) that have been reported in the literature[3,13].

**Computer simulation of piezoelectricity in PVDF.** To support the mechanism proposed in Fig. 4, a full atomistic MD simulation was carried out, following a previous study (for details see Supplementary Note 11 in the Supplementary Information)[29]. As shown in Fig. 5a, a slab of PVDF chains (with 30 repeat units, 12 chains along $x$, and 12 chains along $y$) is sandwiched between two rigid walls (representing crystals), which have positively aligned -$CH_2CF_2$- dipoles along the $y$-direction. The initial slab thickness is set at 6.8 nm (i.e., the strain along $z$, $S_z = 0$). When the $ab$ unit

cell dimensions are set in the same manner as those of the β crystal ($a = 0.858$ nm and $b = 0.491$ nm), no noticeable dipole moment relaxation can be observed within the simulation time period of 2 ns. To reduce interchain interactions and shorten the simulation time, we chose enlarged dimensions for the crystal $ab$ unit cell: $a = 2.1$ nm and $b = 1.2$ nm. After equilibration, we obtained the initial morphology as shown in Fig. 5a ($S_z = 0\%$). Due to the boundary constraint by the crystal walls, there is a finite OAF layer with upward dipoles ($t_{OAF,up} = 0.35$ nm) at each wall with the dipole moments upward. As shown in Fig. 4a, b, upon increasing $S_z$ up to 30%, the $t_{OAF,up}$ gradually increases to 1.75 nm. Due to the increased $t_{OAF,up}$, the polarization along the $y$ direction ($P_y$) near the crystal wall should also increase. Indeed, the $P_y$ values for the first 4 and 8 repeat units from the wall increase with increasing $S_z$ (Fig. 5c). Although it is not possible to quantify the strain-induced $\Delta P$ because of the enlarged $ab$ dimensions, the results of this simulation do provide a qualitative insight into the mechanism for PVDF piezoelectricity, as proposed in Fig. 4.

In summary, this study has provided insight into the fundamental piezoelectric mechanism in BOPVDF in a sample with highly aligned β crystals. The OAF (at least 0.25), which connects between the crystal and the IAF, was deduced from the high $P_s$ (~140 mC/m²). It is the highly mobile OAF that makes possible a high piezoelectric performance (the maximum $d_{33} = -62$ pC/N for a dynamic stress above 0.8 MPa). Computer simulation was used to gain insight into the origin of the negative piezoelectric effect in $d_{33}$, which was found to be explicable in terms of the mechano-electrical interaction between the lamellar crystal and the IAF via the mobile OAF. This understanding enables us to explore even higher piezoelectric performance for P (VDF-TrFE) random copolymers beyond the recently proposed MPB mechanism[3,5].

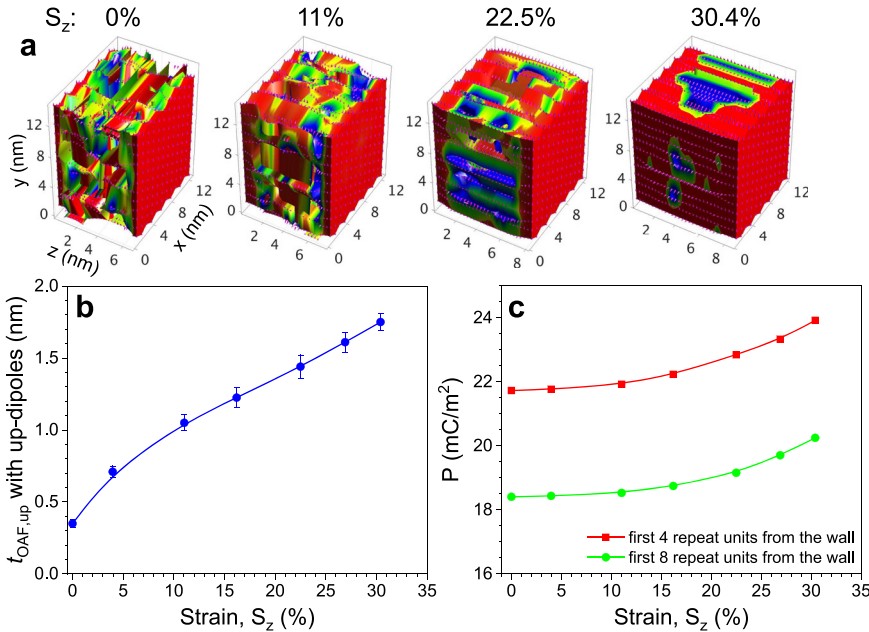

**Fig. 5 Computer simulation of the direct piezoelectricity in PVDF. a** The color-coded simulation slab for PVDF after equilibration at 300 K for 2 ns. The color scale represents PVDF units with positive dipole moments along the *y*-axis with values between zero (blue color) and 2.1 D (red color). The empty space in the pictures is comprised of negative dipoles. There are 12 chains along the *x*-axis, 12 chains along the *y*-axis, and 30 repeat units along the *z*-axis. The slab thickness along *z* is 6.8 nm. Chain ends are attached to both slab walls with a rigid C–C bond (dipole moment fixed along the *y*-axis). The attachment points are organized in the same manner as the β crystal. The *ab* unit cell dimensions on the slab wall are $a = 2.1$ nm and $b = 1.2$ nm, respectively, which are about twice larger than the actual unit cell dimensions of the β crystal. The strain along *z* ($S_z$) is 0%, 11%, 22.5%, and 30.4%, respectively. **b** The $t_{OAF,up}$ and **c** polarization along *y* ($P_y$) for the first 4 and 8 repeat units from the wall as a function of $S_z$.

## Methods

**Materials**. The BOPVDF film with a thickness of 8.0 μm was obtained from Kureha Corporation (Tokyo, Japan). It was fabricated by a biaxial orientation process with stretching along the machine direction (MD) followed by that along the transverse direction (TD)[30]. The stretching ratio along the MD was higher than that along the TD. According to our previous study[17], it has a crystallinity of ca. 0.54 (weight fraction), and the crystals contain 70% α and 30% β phases. The degree of crystal orientation was higher in the MD than in the TD of the biaxially oriented film.

To achieve pure β crystals, the fresh BOPVDF film with an electrode area of 8.04 mm² was unidirectionally poled at 650 MV/m [10 Hz with a DC (325 MV/m) + AC (325 MV/m) unipolar waveform] for at least 40 cycles. Note that if a larger electrode area (e.g., 78.5 mm²) was used, the sample was liable to break down at such a high poling field and only about 10% of the samples would survive. Note that repeated bipolar poling generated a large amount of heat due to dipole/domain flipping, resulting in a much lower breakdown strength (310 ± 16 MV/m peak field, 60 Hz)[31]. Therefore, repeated high-field (>500 MV/m) bipolar poling should be avoided. Unipolar poling, on the other hand, does not involve back and forth ferroelectric switching in BOPVDF, and thus endows the sample with a higher breakdown strength (770 ± 83 MV/m)[31]. Meanwhile, unipolar poling could also align all ferroelectric domains in the normal direction (ND) of the film for later piezoelectric study.

**Characterization and instrumentation**. Two-dimensional (2D) WAXD and SAXS experiments were carried out at the 11-BM Complex Materials Scattering (CMS) beamline at the National Synchrotron Light Source II (NSLS-II), Brookhaven National Laboratory (BNL). The X-ray wavelength (λ) was 0.09183 nm. An in-vacuum CCD (Photonic Science, St. Etienne de St. Geoirs, Isere, France) detector was used for the WAXD experiments. To allow the SAXS signal to pass through, the WAXD detector was shifted and rotated by ~71° from the incident X-ray beam direction[32]. The distance between the sample and the WAXD detector was 224.6 mm, which was calibrated using silver behenate with the first-order reflection at a scattering vector of $q = 1.067$ nm⁻¹, where $q = (4\pi \sin\theta)/\lambda$ with $\theta$ being the scattering half-angle. 1D WAXD curves were obtained by azimuthal integration of the corresponding 2D WAXD patterns. A Pilatus 2 M detector (Dectris, Baden-Dättwil, Switzerland) was used for 2D SAXS experiments. The sample-to-detector distance was 2987.5 mm. 1D SAXS curves were also obtained by azimuthal integration of the corresponding 2D SAXS patterns.

FTIR spectra were collected using a Nicolet iS50R FTIR spectrometer (Thermo Fisher Scientific, Waltham, MA) in a transmission mode at room temperature. The scan resolution was 4 cm⁻¹ with 32 scans.

The thermal behavior of the fresh BOPVDF film was studied by DSC using a TA Instruments (Discovery DSC 250, New Castle, DE, USA). The instrument was calibrated with indium and tin standards. Experiments were carried out in a nitrogen atmosphere using ~5 mg samples crimpled in aluminum pans. The samples were heated from −80 to 200 °C at a heating rate of 10 °C/min.

BDS measurement was carried out on a Novocontrol Concept 80 broadband dielectric spectrometer (Montabaur, Germany) with temperature control. The applied voltage was 1.0 $V_{rms}$ (i.e., root-mean-square voltage) with frequency ranging from 0.01 Hz to 10 MHz and temperature from −100 to 120 °C. Gold (Au) electrodes with an area of 8.04 mm² were evaporated on both surfaces of the BOPVDF film using a Q300TD sputter coater (Quorum Technologies, Ltd., UK). The Au electrode thickness was ~10–15 nm.

The D-E loop measurements were carried out using a Premiere II ferroelectric tester (Radiant Technologies, Inc., Albuquerque, NM) in combination with a Trek 10/10B-HS high-voltage amplifier (0–10 kV AC, Lockport, NY, USA). The applied voltage had either a bipolar or a unipolar sinusoidal waveform at 10 Hz. The film samples were also coated with Au electrodes with an area of 8.04 mm². The Au coating thickness was ~15 nm.

To measure the compression modulus along the ND (i.e., the 3 direction) of the BOPVDF film, a nano-indentation measurement was performed at a constant strain rate of 0.05 s⁻¹, with a maximum indentation depth of 1 μm using a Nano Indenter G200 (Agilent Technologies, USA). All the experiments were conducted by using a Berkovich-shaped diamond indenter. A series of 10 indentations on each sample probe was made for better statistics. The percent to unload, peak hold time, and surface approach velocity of the probe were 90%, 10 s, and 10 nm/s, respectively. The elastic modulus of the specimen was determined from the slope of the load-displacement curve during each unloading cycle[28].

To measure the Young's moduli along the MD (i.e., the 1 direction) and the TD (i.e., the 2 direction) of the BOPVDF film, tensile experiments were measured on a universal test instrument (Model 5965, Instron Instruments, USA) at a cross-head speed of 10 mm/min. The specimen had a dimension of 60 mm in length, 6 mm in width, and 8 μm in thickness. At least three samples were tested for each direction to obtain consistency.

**Piezoelectric measurements**. The experimental setup for $d_{33}$ measurement is given in Supplementary Fig. 6. The test specimen was fixed by two clamps with a mounted glass slide. The $d_{33}$ was measured by applying a weight (50–1000 g) manually on and off the Au-coated poled BOPVDF film (the overlapping area for both top and bottom Au electrodes was 8.04 mm²) through a metal (aluminum or Al) rod as fast as possible. The metal rod (108 g) was always placed on top of the sample to keep the film in right position and make sure it was flat. The generated

piezoelectric voltage and charge were monitored using a Keithley electrometer (model 617, Beaverton, OR, USA), which had an internal impedance >200 TΩ. Through a data-acquisition card (NI USB-6002, National Instruments, USA), the signals were finally recorded with a desktop computer using the LabView software. The Al rod was guided by a home-made fixture to make sure it was in a stable state so that the force could be effectively transferred (i.e., without any friction) onto the top surface of the sample. To make sure the external force was evenly applied to the electroded sample, an Au-coated Cu disk with the same area as the Au electrode was placed between the sample and the Al rod. This Au-coated Cu disk also minimized the triboelectric charge generation induced by loading and unloading the weight on the Al rod, because the triboelectric effect is the smallest between two identical metallic materials, such as Au. A quartz force sensor (model 208C01, PCB Piezotronics, Depew, NY, USA) was installed on the bottom of the metal rod to monitor the force applied to the sample. The direct piezoelectric coefficient $d_{33}$ was calculated by the equation:

$$d_{33} = (Q/A_1)/(F/A_2) \tag{4}$$

where $Q$ is the measured charge, $F$ is the applied force, $A_1$ is the area of the Au electrode on the poled sample, and $A_2$ is the cross-section area where the force was applied. Note that $A_1 = A_2$ for the $d_{33}$ measurement, and then Eq. 4 is simplified as

$$d_{33} = Q/F \tag{5}$$

Considering that $Q = CU$ and $C = \varepsilon_r \varepsilon_0 A_1/d$, $d_{33}$ could also be calculated as

$$d_{33} = \varepsilon_r \varepsilon_0 A_2 U/dF \tag{6}$$

where $C$ is the capacitance of the sample, $U$ is the voltage, and $d$ is the film thickness (8.0 μm) of the poled sample. In addition to this direct piezoelectric charge method, a $d_{33}$ piezo meter (PKD3-2000, PolyK Technologies, State College, PA, USA) using the quasi-static Berlincourt method was also used. The $d_{33}$ results from both methods were compared; see Supplementary Note 9 in the Supplementary Information.

For the $d_{31}$ and $d_{32}$ measurement, long thin strips (30 mm × 5–6 mm) of the BOPVDF film were cut along the 1 and 2 directions, respectively. After coating the strip center with Au electrodes (8.04 mm²) on both surfaces, the sample was unidirectionally polarized at 650 MV/m for 40 cycles. The experimental setup is shown in Supplementary Fig. 7. At the end of the sample, a series of weights ranging from 10 to 200 g were attached while the other end was fixed by a rigid mount to the force sensor. A dynamic force was generated by quickly lifting up the weight, after it was stably hung at the lower end of the film. The output charge induced on the Au electrodes by the dynamic force was measured using the Keithley 617 electrometer and the NI card. The $d_{31}$ and $d_{32}$ could be also calculated as the following:

$$d_{31} \text{ or } d_{32} = (Q/A_1)/(F/A_2) \tag{7}$$

where $A_1 = 8.04$ mm² and $A_2 = 0.040 - 0.048$ mm².

## Data availability

The authors declare that all data supporting the findings of this study are available within the paper and its Supplementary Information file or from the corresponding author upon reasonable request.

## Code availability

The computer code used to simulate the direct piezoelectricity is available upon request to the corresponding author.

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

## Acknowledgements

L.Z. acknowledges partial financial support for the SAXS and WAXD measurements from National Science Foundation, Division of Materials Research, Polymers Program

(DMR-1708990). E.A. acknowledges the support by the Deutsche For-schungsgemeinschaft (DFG) through Grant AL 2058/1-1. Y.H. acknowledge the support by the China Postdoctoral Science Foundation (2019M660035) and Guangdong Basic and Applied Basic Research Foundation (2019A1515110446). This research used the 11-BM CMS beamline of National Synchrotron Light Source-II (NSLS-II), Brookhaven National Laboratory (BNL), a U.S. Department of Energy User Facility operated for the Office of Science by BNL under Contract DE-SC0012704. The authors also thank Dr. Ci Zhang at Case Western Reserve University for AFM measurements.

## Author contributions

L.Z. conceived the idea and supervised the project. L.Z. and Y.H. designed the experiments and wrote the paper. Y.H. conducted the FTIR, WAXD, SAXS, D-E loop, BDS, tensile properties, nanoindentation, and direct $d_{33}$, $d_{32}$, and $d_{31}$ measurements. G.R. conducted D-E loop and BDS measurements, and determined the permanent $P_{r0}$. Q.L. conducted the TEM measurements. E.A. and P.L.T. performed the computer simulation. R.L. and M.F. helped the WAXD and SAXS measurements, and performed data treatment. J.Z.X., G.J.Z., and Z.M.L. participated in the conception of the idea and helped revision of the manuscript.

## Competing interests

The authors declare no competing interests.
