## [Peer Review File · Nature Communications]

REVIEWER COMMENTS

Reviewer #1 (Remarks to the Author):

The authors measured the piezoelectric response of BOPVDF and tried to explore the mechanism of piezoelectric effect of PVDF-based polymers based on their experimental results. It is a fundamental issue that is important for the understanding of the origin of the piezoelectric effect in polymers. They found after a high electric field poling process, the structure of crystalline phase is converted from a mixed phases into more complete β phase. From the dielectric and piezoelectric measurements, they believe part of the amorphous phase (the "OAF" indicated by the authors) contributes to the measured responses. I agree with the authors the amorphous phase may have contribution to the dielectric and piezoelectric response. One of the most important results of this work is that due to the enhanced remnant polarization of the poled BOPVDF, a d_{33} of 62 pC/N is achieved. The piezoelectric response is indeed very large compared with other piezoelectric polymers. However, what I am concerned is whether the authors have correctly measure the piezoelectric response. The high piezoelectric response d_{33} is abruptly increased when the stress is >0.5 MPa. Below that stress, the d_{33} is quite normal. For d_{31} and d_{32} , this abrupt change cannot be observed, which is quite unusual. I suspect that the large d_{33} is caused by the way how the authors measure the d_{33} . They used metal plates to place on the polymer film to avoid triboelectric charge and this might be reason causing the problem: the stress might not be uniformly applied to the film as expected by the authors because the poled film might not be flat. I hope the authors could clarify this issue in the revised manuscript. One method is to use a commercial d_{33} to measure the d_{33} and to find out whether the d_{33} is increased in a similar trend as that reported in this work by changing the static force (simply by tightening the screw of the d_{33} meter). Also, the authors could measure the d_{33} of other poled PVDF-based polymers, for example the BOPVDF poled under an electric field that can't convert the crystal structure from α phase to β phase, to find out if the d_{33} changes in the same way as that reported in the work. Once the correctness of the d_{33} is confirmed, I think the work can be publishable in Nature Communications.

Other issues I hope the authors to note in the revised manuscript:

- (1) The authors put in a lot of work to obtain a real P_r by subtracting the linear part of polarization and polarization relaxed. Because the P_r is not used for quantification, is an exact number of P_r necessary? To improve the readability of the work, I also hope the authors could provide a brief introduction of the techniques and principles to subtract conductivity, linear polarization and relaxed polarization in SI. Otherwise, the readers have to read the lengthy references provided by authors.
- (2) Equation 2 seems not reasonable. Because different dielectric properties of the crystalline phase and amorphous phase and amorphous and crystalline phases form a complex microstructure, the total polarization might not be the linear addition of the polarization of two phases. The authors could check the validity of the equation.

Reviewer #2 (Remarks to the Author):

In this work, authors have studied the origin of the piezoelectricity and its aspects in the case of biaxially oriented (BO) PVDF. They have tried to explore the understanding between the interaction between crystalline and the amorphous regions within BOPVDF films. They deduced that a significant amount (at least 0.25) of an oriented amorphous fraction (OAF) must be present between these two phases. They also provide the theoretical studies performed via Molecular Dynamics simulation, which shows coherence with the experimental findings. Though this is an insightful attempt to be published, some issues need to be addressed before acceptance. The following comments are as follows.

1. The authors should compare the d_{33} values of state-of-the-art values and commercial piezoelectric

polymers, along with advantages and disadvantages.

2. Is γ -PVDF also present in the sample? Also, what is the crystalline content from the FTIR study (Fig. 1b).

3. The manuscript has not shown the imaging of the films. The authors should include the FESEM and AFM images to understand the phenomenon clearly.

4. As the main idea of the manuscript is to depict the interface behavior using oriented amorphous fraction (OAF) and a mobile amorphous fraction (MAF). The PFM results at different locations are also essential to know the local properties.

5. The authors have not commented on dielectric losses, as shown in Fig. S4B,C. Furthermore, at 25 °C, the losses are quite high. What are the ways to minimize the loss?

6. Describe the poling condition as the poling field 650 MV/m is very high.

7. It is outlined in the "Materials" section of the manuscript, that unidirectional poling brings considerable breakdown strength in BOPVDF films. Then, how does the application of sinusoidal waveform with 10 Hz frequency represent unidirectional poling?

8. Grammatical errors: Page 5,6... Other errors: The FTIR figure (Fig. 1b) is mentioned as Fig. 1a. Authors are advised to go through the whole manuscript thoroughly and correct such errors in the manuscript.

Re: Resubmission requested for manuscript ID NCOMMS

Thank you very much for giving us a chance to revise our manuscript to improve its quality. We have purchased a d_{33} piezo meter and performed necessary AFM and TEM experiments in order to address all the review comments. Revisions for Reviewer #1 and Reviewer #2 are highlighted in yellow and cyan, respectively, in the revised manuscript and Supplementary Information (SI).

Reviewer #1:

The authors measured the piezoelectric response of BOPVDF and tried to explore the mechanism of piezoelectric effect of PVDF-based polymers based on their experimental results. It is a fundamental issue that is important for the understanding of the origin of the piezoelectric effect in polymers. They found after a high electric field poling process, the structure of crystalline phase is converted from a mixed phases into more complete β phase. From the dielectric and piezoelectric measurements, they believe part of the amorphous phase (the "OAF" indicated by the authors) contributes to the measured responses. I agree with the authors the amorphous phase may have contribution to the dielectric and piezoelectric response. One of the most important results of this work is that due to the enhanced remnant polarization of the poled BOPVDF, a d_{33} of 62 pC/N is achieved. The piezoelectric response is indeed very large compared with other piezoelectric polymers. However, what I am concerned is whether the authors have correctly measure the piezoelectric response. The high piezoelectric response d_{33} is abruptly increased when the stress is >0.5 MPa. Below that stress, the d_{33} is quite normal. For d_{31} and d_{32} , this abrupt change cannot be observed, which is quite unusual. I suspect that the large d_{33} is caused by the way how the authors measure the d_{33} . They used metal plates to place on the polymer film to avoid triboelectric charge and this might be reason causing the problem: the stress might not be uniformly applied to the film as expected by the authors because the poled film might not be flat. I hope the authors could clarify this issue in the revised manuscript. One method is to use a commercial d_{33} to measure the d_{33} and to find out whether the d_{33} is increased in a similar trend as that reported in this work by changing the static force (simply by tightening the screw of the d_{33} meter). Also, the authors could measure the d_{33} of other poled PVDF-based polymers, for example the BOPVDF poled under an electric field that can't convert the crystal structure from α phase to β phase, to find out if the d_{33} changes in the same way as that reported in the work. Once the correctness of the d_{33} is confirmed, I think the work can be publishable in Nature Communications.

Response:

We thank the reviewer for the overall comment on our work. Following the reviewer’s suggestion, we have purchased a commercial d_{33} meter (PKD3-2000, PolyK Technologies, State College, PA, U.S.A.) and tried to compare the results. Meanwhile, we have thoroughly studied the working mechanism of the quasi-static Berlincourt method for the d_{33} piezo meter from refs. R1-R3 (listed at the end of this response letter).

Fig. R1. (a) Schematic of the d_{33} piezo meter using the quasi-static Berlincourt method. (b) The static (F_{static}) and dynamic force ($F_{dynamic}$) profiles applied to the sample. Usually, F_{static} should be slightly larger than $F_{dynamic}$ to avoid negative total force. The typical frequency is 100 Hz during measurement. (c) d_{33} of the poled BOPVDF as a function of F_{static} with a fixed $F_{dynamic}$ amplitude (0.25 N), measured by our d_{33} piezo meter. (d) d_{33} of the commercial piezoelectric PVDF film (120 μm , PolyK) as a function of $F_{dynamic}$, measured by our direct piezoelectric method. The d_{33} value measured by the d_{33} piezo meter (the red ellipsoid) is shown for comparison. The difference between two methods is around 10%.

Fig. R1a shows the working mechanism of the Berlincourt method. At the very bottom of the piezo meter, there is a reference quartz sensor, on top of which we have a piezo-actuator to generate the dynamic force [$F_{dynamic}(t)$ with a sinusoidal waveform]. For most commercial d_{33} piezo meters, the amplitude of $F_{dynamic}(t)$ ($F_{dynamic}$) is fixed, and it is 0.25 N for our piezo meter (Fig. R1b). The piezo-actuator is driven by an HV power supply with a frequency generator at 100 Hz. On the top of the piezo-actuator, the sample is sandwiched by two metal electrodes. On the very top, we have

a static force (F_{static}) sensor, and the static force is applied by tightening the top screw, and is used to make sure the sample does not rattle. For our d_{33} piezo meter, the F_{static} can be adjusted from 0 to ~ 6.0 N. Since the F_{dynamic} is 0.25 N, the minimum F_{static} had better to be greater than 0.25 N to avoid a negative total force. Otherwise, unreliable results are obtained at low F_{static} .

The reference quartz sensor is used as a force sensor to determine the F_{dynamic} . The amplitude of charge on the sample (Q_{sample}) is measured by the signal detector. Then, the piezoelectric coefficient, d_{33} , can be directly calculated using the piezoelectricity definition:

$$P_{\text{sample}} = Q_{\text{sample}}/A = d_{33}T_3 = d_{33}(F_{\text{dynamic}}/A) \quad (\text{R1})$$

where P_{sample} is the polarization of the sample, T_3 is the dynamic stress, and A is the area of the electrode/sample contact. Note, that the F_{static} does not appear in Eqn. R1; however, it will affect the measured d_{33} results, as we will discuss it later. Since the area A is cancelled on both sides, the d_{33} can be simply calculated using the following equation:

$$d_{33} = Q_{\text{sample}}/F_{\text{dynamic}} \quad (\text{R2})$$

Because A does not appear in Eqn. R2, we do not need to measure the electrode/sample contact area to obtain the T_3 . This is why we do not require an accurate probe geometry for the piezo meter, as long as the top and bottom contact areas of the sample are the same. Also, we do not need to coat both sides of the sample with metal electrodes (if we coat the sample with metal electrodes, it is still fine). This is the advantage of this quasi-static Berlincourt method, and it is really simple to use (i.e., without coating metal electrodes).

Although the F_{static} does not appear in Eqn. R2, it can still affect the measured d_{33} result. For the piezoelectric effect, the maximum stress-induced polarization should be a fixed value. For example, the maximum piezoelectric polarization is determined by the maximum conformational changes of the OAFs in the semicrystalline ferroelectric polymer. In other words, there should be a saturation piezoelectric polarization at a high enough total stress, which includes both static and dynamic stresses. As a result, the higher the applied F_{static} (especially when it approaches the saturation stress), the lower can Q_{sample} and thus d_{33} be achieved. This is exactly seen for our poled BOPVDF film in Fig. R1c and the commercial piezoelectric PVDF film from PolyK.^{R4} This F_{static} effect is also observed for soft piezoelectric ceramics such as PC 5H (PZT).^{R3}

However, the F_{dynamic} effect is just the opposite. Namely, when the F_{static} is kept reasonably low, the larger the F_{dynamic} the higher the measured d_{33} , because the higher F_{dynamic} will induce more polarization and thus the larger piezoelectric effect. This is exactly seen for our experimental results in Fig. 3 in the main text. Again, this is true not only for piezoelectric polymers, but also for piezoelectric ceramics such as PZT (see Fig. 5 of ref. R2).

Unfortunately, most commercial d_{33} piezo meters cannot vary F_{dynamic} . Therefore, we cannot verify the F_{dynamic} effect on d_{33} in Fig. 3a using the purchased d_{33} piezo meter. It is worth noting that in our experiment, even though the force was generated by lifting different weights on the sample,

the force change is very fast, and it is actually a dynamic force, not a static one. To make it clearer, we replaced the “Stress” in Fig. 3 in the main text by “Dynamic stress”. Note: we believe reviewer may have misspoken in the comment. He/she intended to say that we should change the F_{dynamic} (not the F_{static}) to verify our result in Fig. 3a.

In our measurement, an aluminum rod (see Supplementary Fig. 6 in the SI), which is used to transfer the force, is always put on top of the sample. Such a pre-applied weight (108 g) restrained the film in desired position, and kept the film flat to avoid the triboelectricity as much as possible, just like the pre-load static force in commercial d_{33} meter. The corresponding description about such a pre-applied force is added in the revised manuscript. Meanwhile, this F_{static} is small enough that it does not affect the accuracy of the measured d_{33} .

Fig. R2. Room temperature 1D WAXD profiles for a less polarized BOPVDF film (unipolar poling at 500 MV/m for 100 times), when the X-ray is directed along (a) the TD and (b) the ND, respectively. The insets show the corresponding 2D WAXD patterns (in a logarithmic scale). The $(111)_{\text{Au}}$ reflection comes from the residual Au electrodes. (c) Transmission FTIR spectrum for the less polarized BOPVDF film at room temperature. The absorption bands for α and β crystals are labeled.

Fig. R3. Direct piezoelectric charge measurements of (a) d_{33} , (b) d_{32} , and (c) d_{31} for the less polarized BOPVDF film (500 MV/m for 100 times). Using Eqns. 4 and 7 in Methods in the main text, direct piezoelectric coefficients are determined: (d) d_{33} , (e) d_{32} , and (f) d_{31} as a function of the dynamic stress.

Since the commercial d_{33} meter cannot verify the F_{dynamic} -dependent d_{33} , we followed the reviewer's second suggestion. Namely, a less polarized BOPVDF film was obtained by unipolar poling at 500 MV/m for 100 times. Mixed α and β phases were obtained (Fig. R2). Using the FTIR analysis (Fig. R2c), the α and β contents are revealed to be 10.4% and 89.6%, respectively. The piezoelectric properties are shown in Fig. R3, and a lower d_{33} is obtained (Fig. R3a,b). For d_{33} , an abrupt increase is observed between 0.1 and 0.3 MPa, which is broadly consistent with the results for the highly poled BOPVDF in Fig. 3a in the main text. In addition to d_{33} , we also measured d_{32} and d_{31} , as shown in Fig. R3c,d and R3e,f. In concordance with the results reported in our manuscript, d_{32} and d_{31} did not show any abrupt change as function of the dynamic stress.

The different dynamic stress dependences between d_{33} and d_{31}/d_{32} can be explained by the dimensional effect. As described in ref. R5, the dimensional effect is much more significant for d_{33} than d_{31}/d_{32} , because the interchain distance is much more easily changed by an external stress than the covalently bonded chain length. As a result, it is likely that d_{33} has a stronger dependence on the dynamic stress than d_{31}/d_{32} , which is observed in our experimental results in Fig. 3.

Finally, we tested a commercial piezoelectric PVDF film provided by PolyK, using our direct piezoelectric method. The film was uniaxially stretched and corona-poled to achieve a macroscopic dipole moment. The film thickness was 120 μm . As shown in Fig. R1d, the d_{33} is also dependent upon F_{dynamic} and it levels off around 31 pC/N. The d_{33} measured by the piezo meter is between 25 and 28 pC/N, depending on the choice of F_{static} . Since $F_{\text{dynamic}} = 0.25 \text{ N}$ and the electrode area $A = 67.9 \text{ mm}^2$, the dynamic stress T_3 is estimated to be $\sim 3.7 \text{ kPa}$. Comparing

the d_{33} values obtained from these two methods, the difference is found to be about 10%. Therefore, we consider our measurement should be reliable.

The above discussion has been added as Supplementary Note 9 in the SI.

Other issues I hope the authors to note in the revised manuscript:

(1) The authors put in a lot of work to obtain a real P_r by subtracting the linear part of polarization and polarization relaxed. Because the P_r is not used for quantification, is an exact number of P_r necessary? To improve the readability of the work, I also hope the authors could provide a brief introduction of the techniques and principles to subtract conductivity, linear polarization and relaxed polarization in SI. Otherwise, the readers have to read the lengthy references provided by authors.

Response:

First of all, from Eqn. 1 in the main text, the piezoelectric performance is directly related to the permanent P_{r0} in the sample, rather than the P_r during ferroelectric switching in D-E loops. Therefore, we could not use the in-situ P_r during ferroelectric switching to judge the piezoelectric performance. We need to determine the P_{r0} for this study, and the method is similar to the positive-up and negative-down (PUND) method.^{R6}

Following the reviewer's suggestion about the readability, we added the description for subtracting AC electronic conductivity, linear polarization, and measurement of the relaxed polarization in Supplementary Notes 2, 3, 5 of the SI.

To make the subtraction of relaxed polarization clearer, we made the following modifications in the revised manuscript and SI:

“To determine the P_{r0} in the highly poled BOPVDF at different temperatures, we designed the following experiment.” was replaced by *“To subtract the relaxed polarization from the in-situ P_r and determine the P_{r0} in the highly poled BOPVDF at different temperatures, we designed the following experiment.”* on page 9 in the revised manuscript.

“From these D-E loops, P_{r0} , P_r , and E_c were determined (see Supplementary Fig. 5)” was replaced by *“From these D-E loops, P_{r0} , P_r , and E_c were determined according to the method described in Supplementary Note 5 (see Supplementary Fig. 5)”* on page 9 in the revised manuscript.

The title for Supplementary Note 5 in the SI, *“Determination of Permanent Remanent Polarization P_{r0} ”* was replaced by *“Determination of Permanent Remanent Polarization P_{r0} by Subtract the “Relaxed Polarization” (P_r^u)”*.

(2) Equation 2 seems not reasonable. Because different dielectric properties of the crystalline phase and amorphous phase and amorphous and crystalline phases form a complex microstructure, the total polarization might not be the linear addition of the polarization of two phases. The authors could check the validity of the equation.

Response:

Eqn. 2 is based on the definition of bulk polarization P , which is dipole moment per unit volume (see ref. R7). If we assume the total volume does not change, then the bulk P is linearly additive. If we assume three components for the semicrystalline polymer, i.e., crystal, OAF, and IAF, then their polarizations can linearly add together to be the total polarization, P_{total} :

$$P_{\text{total}} = P_{\text{cryst}} + P_{\text{OAF}} + P_{\text{IAF}} \quad (\text{R3})$$

Because IAF does not exhibit any ferroelectricity at room temperature, the nonlinear polarization of the film (P_{NL} or $P_{\text{s,film}}$) will only have contributions from the ferroelectric crystal and OAF:

$$P_{\text{NL}} = P_{\text{s,film}} = P_{\text{cryst}} + P_{\text{OAF}} \quad (\text{R4})$$

Here, $P_{\text{cryst}} = P_{\text{s},\beta\beta}$ and $P_{\text{OAF}} = P_{\text{s},\text{OAF}f\text{OAF}}$. As a result, we obtain Eqn. 2 in the main text:

$$P_{\text{s,film}} = P_{\text{s},\beta\beta} + P_{\text{s},\text{OAF}f\text{OAF}} \quad (\text{R5})$$

In the expectation that this derivation satisfactorily answers the reviewer's question, no particular revision was made in response to this comment.

Reviewer #2:

In this work, authors have studied the origin of the piezoelectricity and its aspects in the case of biaxially oriented (BO) PVDF. They have tried to explore the understanding between the interaction between crystalline and the amorphous regions within BOPVDF films. They deduced that a significant amount (at least 0.25) of an oriented amorphous fraction (OAF) must be present between these two phases. They also provide the theoretical studies performed via Molecular Dynamics simulation, which shows coherence with the experimental findings. Though this is an insightful attempt to be published, some issues need to be addressed before acceptance. The following comments are as follows.

1. The authors should compare the d_{33} values of state-of-the-art values and commercial piezoelectric polymers, along with advantages and disadvantages.

Response:

Actually, we have compared our d_{33} value with the values reported in literature in the original manuscript. On page 6, we wrote the following:

“Above 0.8 MPa, $|d_{33}|$ reached a plateau value around 62 pC/N. This value is significantly higher than those for conventional PVDF, and is similar to that reported for P(VDF-TrFE) copolymer near the 50/50 composition.”

“The maximum values reached were $d_{31} = 22$ pC/N at 41 MPa and $d_{32} = 18$ pC/N at 49 MPa, respectively. These values are typical for conventional PVDF homopolymers.”

The appropriate references are included in the original text.

For the P(VDF-TrFE) 50/50 random copolymer, the high d_{33} is attributed to the morphotropic phase boundary (MPB)-like behavior, namely, crystal conformation transformation from 3/1 helical to all-trans conformation when the VDF content is about 50 mol.%. However, this MPB-like behavior is absent for the PVDF homopolymer. Therefore, a different mechanism must be working, and we attributed it to the stress-induced conformation transformation in the highly mobile OAFs. On the other hand, the P(VDF-TrFE) 50/50 copolymer has a low Curie temperature around 65 °C and is not suitable for high temperature piezoelectric applications.

In addition, we also tested a commercial piezoelectric PVDF (from PolyK), using our direct piezoelectric method; see Response to Comment 1 of the Reviewer #1. As we can see from Fig. R1d, the d_{33} of this commercial PVDF film is 29 pC/N at a dynamic stress of 3.7 kPa. Using the PolyK d_{33} piezo meter, d_{33} was 25-28 pC/N at 3.7 kPa dynamic stress. Apparently, both methods give similar d_{33} values, and this value is significantly lower than that (62 pC/N) of our highly poled BOPVDF film. We have added this in the SI of the revised manuscript.

2. Is γ -PVDF also present in the sample? Also, what is the crystalline content from the FTIR study (Fig. 1b).

Response:

After extensive electric poling, we do not observe any γ -PVDF for the highly poled BOPVDF film. This is also supported by the fact that no $(020)_\gamma$ reflection at 14.3 nm^{-1} and $(120)_\gamma/(022)_\gamma/(112)_\gamma$ reflections at $18\text{-}21 \text{ nm}^{-1}$ are observed in both 1D and 2D WAXD results in Fig. 1a of the main text. For specific γ -PVDF crystalline reflections, please see ref. R8.

3. The manuscript has not shown the imaging of the films. The authors should include the FESEM and AFM images to understand the phenomenon clearly.

Response:

We appreciate the reviewer for this question. Actually, we also would like to visualize this three-phase structure for this BOPVDF, namely, alternating crystal, OAF, and IAF stacks. We first attempted to observe this crystalline morphology of the BOPVDF film by AFM, as shown in Figs. R5a,b. Only micron-sized fibrillar crystals are observed, orienting along the major stretching direction (i.e., the machine direction, MD). No crystalline lamellar stacks could be resolved within the fibrillar crystals due to the resolution limitation of our AFM. Needless to say, we could not observe individual β -crystal (5.78 nm), 1/2 OAF (1.51 nm), and IAF (3.00 nm), as estimated by the SAXS study (see Supplementary Fig. S13 in the SI). Even with a high-resolution AFM and crystalline lamellar stacks visible, the OAF and IAF still could not be seen.^{R9}

In the next effort, we also tried FESEM; however, it does not have the high resolution needed to observe well-defined crystalline lamellar stacks with crystal, OAF and IAF either. Therefore, we abandoned the FESEM approach, and used TEM instead. Note, PVDF is extremely resistant to chemicals, and no gas-phase staining agent (e.g., RuO_4) could stain PVDF, as we reported before.^{R10} One wet-staining method was reported before with extremely harsh chemical conditions.^{R11} Basically, the PVDF sample needs to be immersed in a strongly oxidizing agent

(CrO₃ and P₂O₅ in concentrated H₂SO₄) and reacted for 2-24 h at 90 °C. We tried this method and the entire TEM grid with the microtomed thin sections were destroyed by the strong oxidizing acid. It is also reported that PVDF can react with a strong base, such as KOH, in isopropanol during reflux.^{R12} After reaction, conjugated double bonds can form in the main chain due to dehydrofluorination. Then, it is possible to use RuO₄ to stain the double bonds.^{R13} However, we also failed in this method because all the thin sections fell off the TEM grids after reflux in isopropanol. We consider that wet chemical etching cannot be used, because either the entire TEM grid or the thin sections are so easily destroyed.

Fig. R5. AFM (a) 2D top-view and (b) 3D-view phase images of the fresh BOPVDF film. (c) Bright-field HR-TEM image of highly poled BOPVDF thin section showing the (110/200) crystalline fringes (0.428 nm).

Therefore, we resort to high-resolution TEM (HR-TEM).^{R14,15} Since PVDF is also extremely e-beam stable, its crystalline fringes could be directly observed by HR-TEM. Fig. R5c shows the bright-field HR-TEM image for a cryomicrotomed thin-section (~80 nm obtained at -50 °C) of the poled BOPVDF film. Crystalline fringes are observed with a *d*-spacing of 0.428 nm, which corresponds to the (110/200)_β spacing. Nonetheless, no obvious IAF/1/2 OAF/crystal/1/2 OAF/IAF structure could be seen (e.g., see the red dashed lines in Fig. R5c) We consider that this may be attributed to the relatively thick microtomed section of ~80 nm, and multiple crystalline lamellae are present in the thickness direction. This makes it impossible to clearly see the IAF/1/2 OAF/crystal/1/2 OAF/IAF structure in the TEM projection image. After numerous attempts, we still could not obtain less than 20 nm thin sections by microtoming below the T_g (-40 °C) of PVDF, using a diamond knife. Meanwhile, without a suitable gas-phase staining agent, we also doubt that IAF can differentiate from OAF and crystal. As such, we consider this visualization effort is beyond the scope of this work, although we are dedicated to continue this effort in the future. We trust that the reviewer will understand our situation, and appreciate why no revision has been made to the manuscript.

4. As the main idea of the manuscript is to depict the interface behavior using oriented amorphous fraction (OAF) and a mobile amorphous fraction (MAF). The PFM results at different locations are also essential to know the local properties.

Response:

As we have mentioned above, our AFM could not reach a sufficiently high resolution (below 1 nm) to clearly see IAF (3.00 nm), $\frac{1}{2}$ OAF (1.51 nm), and crystal (5.78 nm). Actually, such a high resolution has been difficult for most AFMs. Therefore, it is impossible for us to achieve viable PFM images and study the polarization behavior of IAF, OAF, and crystal in the BOPVDF film. It is our hope that the reviewer will appreciate this technical difficulty and forgive us for not being able to achieve this goal.

5. The authors have not commented on dielectric losses, as shown in Fig. S4B,C. Furthermore, at 25 °C, the losses are quite high. What are the ways to minimize the loss?

Response:

Yes, it is true. The dielectric loss for the highly poled BOPVDF is indeed higher than the fresh BOPVDF (see Supplementary Fig. 4). This is attributed to the highly poled β crystals with the macroscopic dipole moment in the film normal direction. Because of the large permanent P_{r0} in the poled BOPVDF film, all the nonlinearity harmonics (D_n with $n \geq 2$) substantially increase, including the even-numbered harmonics.^{R16,17} Note, all the nonlinear harmonics contribute to the dielectric loss. This is why the poled BOPVDF film exhibits a high dielectric loss. Given this mechanism, it is difficult to reduce the dielectric loss for the poled BOPVDF film, unless we anneal the sample at 120 °C to reduce the P_{r0} (note that P_{r0} decreases above 80 °C; see Fig. 2f). However, decreasing P_{r0} will decrease the piezoelectric property,^{R18} and this is contrary to our purpose. Finally, we consider that the high dielectric loss should not be a detrimental factor for piezoelectricity, because the poled BOPVDF film exhibited a rather linear D-E loop when the poling field is below 50 MV/m (see Fig. 2d).

Following the reviewer's suggestion, we added some comments on the high dielectric loss in the SI after Supplementary Fig. 4:

“The high dissipation factor for the poled BOPVDF is attributed to the dielectric nonlinearity caused by the high P_{r0} ,^{4,5} and cannot be minimized without decreasing P_{r0} and the piezoelectric property. However, this low-field dissipation will not significantly increase the hysteresis loop loss for piezoelectricity, because the D-E loop is rather linear and narrow when the poling electric field is below 50 MV/m (see Fig. 2d in the main text).”

6. Describe the poling condition as the poling field 650 MV/m is very high.

Response:

To reduce the probability of dielectric breakdown at such a high field, we often used electrode areas smaller than 3.5 mm in diameter. When 10 mm diameter electrodes were used, the survival rate for high-field poling at 650 MV/m was only about 10%. To clarify this issue, we revised the poling condition on page 16 in the revised manuscript:

“To achieve the pure β phase, the fresh BOPVDF film with an electrode area of 8.04 mm² was unidirectionally poled at 650 MV/m [10 Hz with a DC (325 MV/m) + AC (325 MV/m) unipolar waveform] for at least 40 cycles. Note that if a larger electrode area (e.g., 78.5 mm²) was used,

the sample was liable to break down at such a high poling field and only about 10% of the samples would survive.”

7. It is outlined in the “Materials” section of the manuscript, that unidirectional poling brings considerable breakdown strength in BOPVDF films. Then, how does the application of sinusoidal waveform with 10 Hz frequency represent unidirectional poling?

Response:

For bipolar poling, we employed a conventional sinusoidal waveform, as shown in Fig. R6a. The unipolar poling means AC + DC, and the electric field is always along one direction, as shown in Fig. R6b.

To clarify this point, we modified our description in the 2nd paragraph, the “Materials” section of the revised manuscript, as described in response to comment 6 (above), giving the revised text:

“To achieve the pure β phase, the fresh BOPVDF film with an electrode area of 8.04 mm² was unidirectionally poled at 650 MV/m [10 Hz with a DC (325 MV/m) + AC (325 MV/m) unipolar waveform] for at least 40 cycles. Note that if a large electrode area (such as 78.5 mm²) was used, the sample was liable to break down at such a high poling field and only about 10% of the samples would survive.”

Fig. R6. (a) Bipolar (AC) and (b) unipolar (DC+AC) sinusoidal wave functions used in this work.

8. Grammatical errors: Page 5,6... Other errors: The FTIR figure (Fig. 1b) is mentioned as Fig. 1a. Authors are advised to go through the whole manuscript thoroughly and correct such errors in the manuscript.

Response:

Thanks and we have corrected this. Also, we have carefully gone through the entire manuscript, in a vigorous effort to eliminate any grammatical or other errors in the revised manuscript.

Additional Revisions

Finally, we have decided that it would be wise to change the terminology “mobile amorphous fraction (MAF)” to “isotropic amorphous fraction (IAF)”. The reason for this change is that we feel we should make apparent the contrast between isotropy and anisotropy, rather than refer to a MAF, whose natural counterpart is a rigid amorphous fraction (RAF). This has been changed throughout all text and figures in this manuscript.

References

- R1. Dragan Damjanovic. Stress and frequency dependence of the direct piezoelectric effect in ferroelectric ceramics. *J. Appl. Phys.* **82**, 1788-1797 (1997).
- R2. Barzegar, A. F., Damjanovic, D. & Setter, N. The effect of boundary conditions and sample aspect ratio on apparent d_{33} piezoelectric coefficient determined by direct quasistatic method. *IEEE Trans. Ultrason. Ferroelectr. Freq. Control* **51**, 262-270 (2004).
- R3. Stewart, M., Cain, M.G., Direct piezoelectric measurement: The Berlincourt method. In: Cain M. (eds.) Characterisation of Ferroelectric Bulk Materials and Thin Films. *Springer Series in Measurement Science and Technology, Vol 2*. (Springer, Dordrecht, 2014).
- R4. PolyK website: <https://piezopvdf.com/static-force-sensor-10N>.
- R5. Tashiro, K., Kobayashi, M., Tadokoro, H. & Fukada, E. Calculation of elastic and piezoelectric constants of polymer crystals by a point-charge model - application to poly(vinylidene fluoride) form-I. *Macromolecules* **13**, 691-698 (1980).
- R6. Feng, S. M., Chai, Y. S., Zhu, J. L., Manivannan, N., Oh, Y. S., Wang, L. J., Yang, Y. S., Jin, C. Q., & Kim, K. H. *New J. Phys.* **12**, 073006 (2010).
- R7. Kao, K. C. Dielectric Phenomena in Solids: with Emphasis on Physical Concepts of Electronic Processes (Elsevier Academic Press, Boston, 2004).
- R8. Andrew J. L. Annealing of poly(vinylidene fluoride) and formation of a fifth phase. *Macromolecules* **15**, 40-44 (1982).
- R9. Schmid, U., Schneider, M., Teuschel, M., Hafner, J. Origin of the strong temperature effect on the piezoelectric response of the ferroelectric (co-)polymer P(VDF₇₀-TrFE₃₀). *Polymer* **170**, 1-6 (2019).
- R10. Huang, H., Chen, X., Li, R., Fukuto, M., Schuele, D. E., Ponting, M., Langhe, D., Baer, E., Zhu, L. Flat-on secondary crystals as effective blocks to reduce ionic conduction loss in polysulfone/poly(vinylidene fluoride) multilayer dielectric films. *Macromolecules* **51**, 5019-5026 (2018).
- R11. Vaughan, A. S. Etching and morphology of poly(vinylidene fluoride). *J. Mater. Sci.* **28**, 1805-1813 (1993).
- R12. Rabuni, M. F., Sulaiman, N. M. N., Aroua, M. K., Hashim, N. A. Effects of alkaline environments at mild conditions on the stability of PVDF membrane: An experimental study. *Ind. Eng. Chem. Res.* **52**, 15874-15882 (2013).
- R13. Trent, J. S., Scheinbeim, J. I., Couchman, P. R. Ruthenium tetroxide staining of polymers for electron microscopy. *Macromolecules* **16**, 589-598 (1983).
- R14. Bai, M., Li, X. Z., Ducharme, S. Electron diffraction study of the structure of vinylidene fluoride-trifluoroethylene copolymer nanocrystals. *J. Phys.: Condens. Matter* **19**, 196211 (2007).

- R15. Lolla, D., Gorse, J., Kisielowski, C., Miao, J., Taylor, P. L., Chase, G. G., Reneker, D. H. Polyvinylidene fluoride molecules in nanofibers, imaged at atomic scale by aberration corrected electron microscopy. *Nanoscale* **8**, 120-128 (2016).
- R16. Furukawa, T., Nakajima, K., Koizumi, T., Date, M. Measurements of nonlinear dielectricity in ferroelectric polymers. *Jpn. J. Appl. Phys.* **26**, 1039-1045 (1987).
- R17. Li, Y., Ho, J., Wang, J., Li, Z.-M., Zhong, G.-J., Zhu, L. Understanding nonlinear dielectric properties in a biaxially oriented poly(vinylidene fluoride) film at both low and high electric fields. *ACS Appl. Mater. Interfaces* **8**, 455-465 (2016).
- R18. Silva, M. P., Costa, C. M., Sencadas, V., Paleo, A. J., Lanceros-Mendez, S. Degradation of the dielectric and piezoelectric response of β -poly(vinylidene fluoride) after temperature annealing. *J. Polym. Res* **18**, 1451-1457 (2011).

We hope we have adequately addressed all reviewers' comments. If there is any way in which I can be of help to the review process, please feel free to contact me.

Sincerely,

REVIEWERS' COMMENTS

Reviewer #1 (Remarks to the Author):

The authors response fully to my questions and concern. I don't have further questions.

Reviewer #2 (Remarks to the Author):

Authors have revised their manuscript as per reviewers' suggestions/comments. I am satisfied with the revision. Manuscript may be accepted as such for publication.